# Extracorporeal Shockwave Therapy (ESWT) Alleviates Pain, Enhances Erectile Function and Improves Quality of Life in Patients with Chronic Prostatitis/Chronic Pelvic Pain Syndrome

**DOI:** 10.3390/jcm10163602

**Published:** 2021-08-16

**Authors:** Wen-Ling Wu, Oluwaseun Adebayo Bamodu, Yuan-Hung Wang, Su-Wei Hu, Kai-Yi Tzou, Chi-Tai Yeh, Chia-Chang Wu

**Affiliations:** 1Department of Urology, Shuang Ho Hospital, Taipei Medical University, New Taipei City 235, Taiwan; 15334@s.tmu.edu.tw (W.-L.W.); 16625@s.tmu.edu.tw (O.A.B.); 10352@s.tmu.edu.tw (S.-W.H.); 11579@s.tmu.edu.tw (K.-Y.T.); 2TMU Research Center of Urology and Kidney (TMU-RCUK), Taipei Medical University, Taipei City 110, Taiwan; 3Department of Hematology and Oncology, Shuang Ho Hospital, Taipei Medical University, New Taipei City 235, Taiwan; 4Department of Medical Research, Shuang Ho Hospital, Taipei Medical University, New Taipei City 235, Taiwan; d508091002@tmu.edu.tw (Y.-H.W.); ctyeh@s.tmu.edu.tw (C.-T.Y.); 5Graduate Institute of Clinical Medicine, College of Medicine, Taipei Medical University, Taipei 110, Taiwan; 6Department of Urology, School of Medicine, College of Medicine, Taipei Medical University, Taipei City 110, Taiwan; 7Department of Medical Laboratory Science and Biotechnology, Yuanpei University of Medical Technology, Hsinchu City 30015, Taiwan

**Keywords:** chronic prostatitis, chronic pelvic pain syndrome, extracorporeal shockwave therapy, ESWT, NIH-CPSI, EHS, IIEF-5, QoL

## Abstract

Purpose: Chronic prostatitis/chronic pelvic pain syndrome (CP/CPPS), affecting over 90% of patients with symptomatic prostatitis, remains a therapeutic challenge and adversely affects patients’ quality of life (QoL). This study probed for likely beneficial effects of ESWT, evaluating its extent and durability. Patients and methods: Standardized indices, namely the pain, urinary, and QoL domains and total score of NIH-CPSI, IIEF-5, EHS, IPSS, and AUA QoL_US were employed in this study of patients with CP/CPPS who had been refractory to other prior treatments (*n* = 215; age range: 32–82 years; median age: 57.5 ± 12.4 years; modal age: 41 years). Results: For CP symptoms, the mean pre-ESWT NIH-CPSI total score of 27.1 ± 6.8 decreased by 31.3–53.6% over 12 months after ESWT. The mean pre-ESWT NIH-CPSI pain (12.5 ± 3.3), urinary (4.98 ± 2.7), and QoL (9.62 ± 2.1) domain scores improved by 2.3-fold, 2.2-fold, and 2.0-fold, respectively, by month 12 post-ESWT. Compared with the baseline IPSS of 13.9 ± 8.41, we recorded 27.1–50.9% amelioration of urinary symptoms during the 12 months post-ESWT. For erectile function, compared to pre-ESWT values, the IIEF-5 also improved by ~1.3-fold by month 12 after ESWT. This was corroborated by EHS of 3.11 ± 0.99, 3.37 ± 0.65, 3.42 ± 0.58, 3.75 ± 0.45, and 3.32 ± 0.85 at baseline, 1, 2, 6, and 12 months post-ESWT. Compared to the mean pre-ESWT QoL score (4.29 ± 1.54), the mean QoL values were 3.26 ± 1.93, 3.45 ± 2.34, 3.25 ± 1.69, and 2.6 ± 1.56 for months 1, 2, 6, and 12 after ESWT, respectively. Conclusions: This study shows ESWT, an outpatient and easy-to-perform, minimally invasive procedure, effectively alleviates pain, improves erectile function, and ameliorates quality of life in patients with refractory CP/CPPS.

## 1. Introduction

Prostatitis affects an estimated 8.2% of the global population and remains a major health issue [1]. Added to the therapeutic challenge it poses to physicians, prostatitis adversely affect patients’ quality of life (QoL) [2] and causes patients substantial economic constraint [3]. The National Institutes of Health (NIH) clinical syndromes-based classification system divides prostatitis into four categories: namely, category I, which includes acute systemic infection and replaces the so-called ‘acute bacterial prostatitis’; category II, which replaces the erstwhile ’chronic bacterial prostatitis’, and comprises recurrent urinary tract infection (UTI) in men with prostatic bacterial presence between infections; category III for chronic prostatitis/chronic pelvic pain syndrome (CP/CPPS), evidenced by chronic pelvic pain with no known alternative attributable pathology; and category IV for asymptomatic prostatitis based on biopsy- or semen analysis-confirmed inflammation [3,4,5].

Protracted painful prostatitis, herein termed CP/CPPS, affects over 90% of patients with symptomatic prostatitis [6], and is characterized by persistent or recurring pain/discomfort in the pelvis for at least 3 of the last 6 months, often accompanied by lower abdominal pain; painful ejaculation; genital pain; lower urinary tract symptoms (LUTS) such as hesitancy, straining, feeling of incomplete bladder emptying, poor or intermittent stream, dribbling, prolonged micturition, urgency, frequency, or nocturia; psycho-social impairments; and erectile/sexual dysfunction [3,4,5,6].

Over the last six decades, CP/CPPS, attributed to infection, inflammation, impaired urothelial integrity and function, endocrine imbalance, autoimmunity, voiding dysfunction, or neuropsychological factors [7,8], has remained a ‘diagnosis of exclusion’ with currently unclear or inexact underlying cause, thus stimulating interest and concerted research effort to demystify its etiology and unravel probable underlying molecular mechanisms. Recently, *Trichomonas Vaginalis* infection has been suggested as a probable pathoetiologic factor in CP/CPPS because of its complicity in chronic persistent prostatic infection and prostate epithelial cell inflammation [9]. Being able to cause inflammation by adhering to normal prostate epithelial cells [9,10], the association of *T. Vaginalis* with benign prostate hyperplasia (BPH) and prostate cancer is also currently being investigated [11,12]. However, the effect of *T. Vaginalis* on the development of chronic prostatitis remains unclear [13,14].

Despite advances in diagnostic and therapeutic approaches based on our evolving understanding of the CP/CPPS etiopathology, there is no international consensus-based approved single agent therapy with proven high efficacy against this syndrome [15], thus, the adoption of multi-modal approaches to treating CP/CPPS [16] such as the ’three As’. The ’three As’ modality consists of α-blockers, antibiotics, and/or anti-inflammatory/immune modulation therapy. There is mounting evidence supporting the therapeutic efficacy of the three As in some patients with CP/CPPS [17]. The magnitude of effect and the disproportional mean decrease in the NIH Chronic Prostatitis Symptom Index (NIH—CPSI) and response rates in treatment groups in comparison to placebo groups suggest the superiority of directed multi-modal therapy over monotherapy, and advocate consideration of these agents for optimal management of patients with CP/CPPS [17]. Alternatively, phytotherapies, including quercetin, Cernilton, Eviprostat/pollen extract, and pentosane polysulfate [17,18], as well as non-pharmacological therapies such as acupuncture and extracorporeal shockwave therapy (ESWT), have also shown some efficacy in the treatment of CP/CPPS [8].

The UPOINTS algorithm, formed by addition of the sexuality (S) component to the original UPOINT system consisting of urinary domain (U), psycho-social (P), organ-specific (O), infection (I), neurological (*n*), and muscle tension and tenderness (T) domains, helps stratify patients into clusters of homogeneous clinical presentation, identifies recognizable phenotypes, and proposes specific treatment plans [19]. Accruing evidence indicates that treatment of patients consistent with this complex multi-modal disease phenotype-based therapeutic approach elicits clinically appreciable amelioration of CP/CPPS symptomatology in many patients, with the addition of second-line therapeutics such as 5-phosphodiesterase inhibitors, antidepressants, muscle relaxants, and anxiolytics to help elicit satisfactory treatment response in patients with sub-optimal response to initial first-line therapy [20]. There are reports associating the UPOINTS algorithm with clinical improvement in 75–84% of CP/CPPS cases [5,19,20,21].

As already mentioned, non-pharmacological therapies are also touted as effective against CP/CPPS [8,22]. ESWT is one such non-pharmacological treatment modality [22]. ESWT is well-known and widely used in urological clinics to treat Peyronie’s disease, erectile dysfunction (ED), and chronic pelvic pain [23]. Zimmermann R. et al. first reported the use of ESWT for treating CP/CPPS in 2009. Their seminal report demonstrated the ease and safety of ESWT, as well as showed that all patients with CP/CPPS completed their treatment without complications and that follow-up was uneventful, with all treated patients exhibiting marked amelioration of pain, improved QoL, and better voiding conditions following ESWT, compared with progressive deterioration in the placebo group [24]. It has been suggested that the observed post-ESWT improvement in CP/CPPS may be due to “reducing passive muscle tone, hyperstimulating nociceptors, interrupting the flow of nerve impulses, or influencing the neuroplasticity of the pain memory” [25].

Despite these touted beneficial effects of ESWT on CP/CPPS, there are suggestions that its therapeutic effects may be short-lived, with tendency to decrease in month 6 of follow-up [23]. However, contradictory results on the effect of ESWT on CP/CPPS abound, especially with a dearth of long-term follow-up. Considering the short duration (3 months) of the premier ESWT study and the unusual lack of placebo response in the control group, as rightly posed by Marszalek M [25], outstanding questions linger regarding (i) suitable patient demographics or selection criteria for the treatment, (ii) the probable potentiating effect of previous treatment strategies, and (iii) the unclear durability of treatment benefit for lack of longer term effect data [23,24,25]. Thus, the present study evaluates the therapeutic effect of ESWT on CP/CPPS patients with prior treatment failure.

## 2. Methods

### 2.1. Patients

This single-center, prospective, single-arm cohort study was performed from September 2016 to January 2018 at the Shuang Ho Hospital, Taipei Medical University, New Taipei, Taiwan. A total of 215 patients with established diagnosis of CP/CPPS, non-inflammatory type (NIH type IIIb prostatitis), were included in our study. The study was approved by Taipei Medical University-Joint Institutional Review Board (Approval No.: N201712069), and written informed consent was obtained from all the enrolled patients. The study protocol was compliant with the Declaration of Helsinki.

Enrolled patients were seen in the outpatient settings. Diagnosis was established after thorough history-taking, physical examination, and screening with the following examinations: (i) urine analysis, (ii) urine culture, (iii) semen analysis, (iv) semen culture, (v) nucleic acid amplification test (NAAT) for *T. Vaginalis*, (vi) NAAT for *Chlamydia trichomatis*, (vii) blood test, including complete blood count/differential count, and C-reactive protein (CRP), (viii) prostate ultrasound, and (ix) kidney, ureter, and bladder (KUB) radiography.

### 2.2. Inclusion and Exclusion Criteria

Inclusion criteria were as follows: patients (i) aged 18 or above, (ii) diagnosed with CP/CPPS, (iii) suffered prostatitis-like symptoms for at least the last 6 months with no identifiable cause, (iv) refractory to administered medical therapies for at least the last 6 months. The exclusion criteria included (i) anatomical abnormalities of the genito-urinary system, (ii) urinary tract or perineal region infection, (iii) cancer of the genito-urinary system, (iv) prostate specific antigen >4, and (v) major surgery of the pelvic organs, including the prostate or rectum.

### 2.3. ESWT Protocol

All patients were treated in the dorsal recumbent position with perineal ESWT once a week for 6 consecutive weeks with a protocol of 3000 pulses at an energy of 0.25 mJoule/mm^2^ and a frequency of 4 Hertz (Hz) using DUOLITH^®^ SD1 (Storz Medical AG, Tägerwilen, Switzerland). Probe position was changed after every 500 pulses to broaden the therapy effect field, induce re-perfusion of the prostate, improve the hemodynamic profile of the prostatic artery, and forestall probable procedure-associated side-effects, such as, itchy or painful dysesthesia, ecchymosis, and petechiae. One cycle consisted of 6 sessions. The DUOLITH^®^ SD1 is a mobile shockwave therapy apparatus with a SEPIA^®^ hand-piece for ease of manipulation and positioning to facilitate focused shock waves.

### 2.4. Evaluation of Outcome

The primary outcomes of the present study, namely, pain reduction and amelioration of urinary symptoms, were evaluated using the NIH-CPSI, International Prostate Symptom Score (IPSS), and American Urological Association Quality of Life due to Urinary Symptoms (AUA QOL_US), while improved sexual function, being the secondary outcome, was assessed using the International Index of Erectile Function (IIEF), and Erection Hardness Score (EHS). All questionnaires were completed after detailed explanation during clinic visits (i) before commencing ESWT, (ii) after the third ESWT session, (iii) a week after the sixth ESWT session, (iv) 1 month, (v) 2 months, (vi) 6 months, and (vii) 12 months after the last ESWT session. Aside from ESWT treatment, all patients with concomitant *T. vaginalis* infection (*n* = 19) were given a single dose of 2 g Metronidazole. None of the enrolled subjects underwent transurethral resection of the prostate (TURP) during follow-up, nor did any receive other therapies concomitantly with ESWT.

### 2.5. Statistical Analyses

All statistical analyses were performed using IBM SPSS Statistics for Windows, Version 25.0 (IBM Corp. Released 2017, Armonk, NY, USA: IBM Corp). For randomly missing data, we used the pairwise deletion (also known as the ‘available case analysis’) by deleting any case with missing variables required for a specific analysis, but including such cases in analyses where all required variables were present. Pearson’s chi-square (χ^2^) test was used to determine the relationship or association between categorical variables. The paired sample t-test was used for comparing two dependent sample means, while the independent t-test was used to compare independent sample means. *p* values ≤ 0.05 were considered statistically significant.

## 3. Results

The present study evaluated the effect of ESWT on pain, erectile function, and QoL in patients with CP/CPPS (*n* = 215) using standardized evaluation indices, namely the pain domain, urinary domain, QoL domain, and total score of NIH-CPSI, IIEF-5, EHS, IPSS, and AUA QoL_US. Participants were aged 32–82 years (mean: 57.1 ± 12.41 years; median: 57.5 ± 12.41 years; modal age: 41 years).

For CP symptoms, the mean NIH-CPSI pain, urinary, and QoL domains, as well as total score before ESWT were 12.53 ± 3.25, 4.98 ± 2.72, 9.62 ± 2.06, and 27.10 ± 6.81, respectively. Compared to these baseline values, the mean NIH-CPSI total scores decreased by 31.3%, 37.3%, 35.7%, and 53.6% at 1, 2, 6, and 12 months after ESWT administration, respectively (Appendix A). Per component, we observed a 2.3-fold, 2.2-fold, and 2.0-fold improvement in the CPSI pain, urinary and QoL domains, respectively, by month 12 post-ESWT (Figure 1; also see Appendix A).

For erectile function, the IIEF-5 also improved significantly after ESWT, as demonstrated by mean IIEF-5 scores of 18.43 ± 6.34 (1.1-fold), 20.42 ± 5.59 (1.3-fold), 20.25 ± 5.94 (1.3-fold), and 18.65 ± 6.85 (1.2-fold) at months 1, 2, 6, and 12 respectively, compared to the mean IIEF-5 score of 15.82 ± 7.70 before ESWT (Appendix A). This was corroborated by the improved EHS of 3.37 ± 0.65, 3.42 ± 0.58, 3.75 ± 0.45, and 3.32 ± 0.85 at 1, 2, 6, and 12 months post-ESWT, respectively, compared to baseline (3.11 ± 0.99) (Figure 2A,B; also see Appendix A).

Consistent with the NIH-CPSI, the severity of LUTS was ameliorated as measured by the IPSS. In comparison to the mean pre-ESWT IPSS of 13.9 ± 8.41, we recorded a 27.1%, 38.0%, 42.0%, and 50.9% time-dependent improvement, respectively, of urinary symptom severity at months 1, 2, 6, and 12 of ESWT (Figure 2C; Also see Appendix A).

Understanding that the severity of urinary symptoms, including pain, affects patients’ QoL, we evaluated and demonstrated commensurate improvement in patients’ QoL as per the AUA QOL_US. The mean QoL score before ESWT was 4.29 ± 1.54. For the first, second, sixth, and twelfth months following ESWT, we recorded mean QoL values of 3.26 ± 1.93, 3.45 ± 2.34, 3.25 ± 1.69, and 2.6 ± 1.56, respectively (Figure 2D; also see Appendix A).

A baseline-normalized paired sample mean of all evaluated parameters is shown in Table 1. Compared to pre-ESWT status, ESWT elicited statistically significant improvement in all patients’ clinical parameters (*p* < 0.001), except for the EHS at 2 months (mean baseline-paired difference = 0.23, *p* = 0.096), 6 months (mean baseline-paired difference = 0.25, *p* = 0.351), and 12 months (mean baseline-paired difference = 0.10, *p* = 0.302) following ESWT, compared to the 40.9% mean improvement in EHS (*p* = 0.009) at 1 month following ESWT (Table 1).

## 4. Discussion

In the past decades, several studies across different medical disciplines have indicated the therapeutic efficacy of ESWT to various degrees against diverse medial conditions, including spasticity after upper motor neuron injury [26], tendinopathies, musculoskeletal conditions and soft tissue disorders [27,28,29,30,31,32], refractory angina pectoris [33], erectile dysfunction [34], and sexual conditions other than erectile dysfunction [35,36]. While several studies have also suggested that the use of ESWT exerts a beneficial effect in patients with CP/CPPS [8,15,16,17,18,19,20,21,22,23,24], as with erectile dysfunction [37], the application of ESWT in the management of CP/CPPS is not without its controversies [23,25].

Although ESWT has been touted as a major therapeutic advance in the field of CP/CPPS in recent decades, as briefly summarized in Table 2, it remains far from being a perfect treatment paradigm and harbors certain limitations as already alluded to earlier [23,24,25].

The present study demonstrated the beneficial effect of ESWT on pain, erectile function, and QoL in patients with CP/CPPS (*n* = 215) at our facility based on improved pain domain, urinary domain, QoL domain, and total score of NIH-CPSI, IIEF-5, EHS, IPSS, and AUA QoL_US. Our findings are consistent with those of Yuan P. et al.’s meta-analysis, which demonstrated that low-intensity ESWT (Li-ESWT) was significantly efficacious in treating patients with CP/CPPS throughout the follow-up of 4 and 12 weeks, as well as at the 24-week endpoint, despite the statistically insignificant effect difference at 24-week follow-up due to insufficient data [38].

In our study, we demonstrated significant alleviation of pain in patients after ESWT. As mentioned by Zimmerman R et al. [24], the observed pain alleviation may be attributed to intracellular alterations following conversion of the mechanical extracorporeal shock-waves to biochemical signals. In addition to enhanced local microvascularization, coupled with reduced residual muscle tension and spasticity [24], we posit that the pulsatile stimulation of pain receptors (nociceptors) by ESWT disrupts in part or completely impedes the transmission of potential pain stimuli; it is also probable that ESWT simply overstimulates the nociceptors beyond their sensitivity threshold with consequent numbing of the sensory neurons to noxious stimuli, thus resulting in reduced pain perception. Concordant with the “neural pain memory” hypothesis put forward by Wess OJ [39], it is also conceivable that due to the plasticity of synapses, ESWT possibly effaces the noxious link established between pain sensory input and motor nerve signal output, and thereby reverses the sensation of chronic pain. Essentially, ESWT elicits the alleviation of pain by selectively eliminating pathological reflex patterns [24,39].

Furthermore, apart from pain alleviation, we also demonstrated that ESWT ameliorated the severity of other prostatitis symptoms in our CP/CPPS cohort with a 53.6% decrease in NIH-CPSI, 17.9% increase in IIEF-5, 6.8% increase in EHS, and 50.9% decrease in IPSS by month 12 after ESWT, concordant with the beneficial effect of ESWT in patients with CPPS (17% decrease in NIH-CPSI, 5.3% increase in IIEF, and 25% decrease in IPSS) reported by Zimmerman R et al. by month 3 after ESWT [24]. Additionally, this is consistent with the conclusions of a recent meta-analysis that “-ESWT showed great efficacy for the treatment of CP/CPPS at the endpoint and during the follow-up of 4 and 12 weeks” [38].

Moreover, because CP/CPPS-pathognomonic ED and LUTS significantly affect QoL, we demonstrated that ESWT improves the QoL of patients with CP/CPPS. This aligns with Zimmermann R et al.’s findings [24], and with reports that over 80% of patients that were non-responsive to therapy responded to ESWT by month 3, thus projecting ESWT as a salvage or rescue treatment for restoring clinical ability and improving QoL in patients with CP/CPPS who were refractory to the traditional ’three As’ therapy [40]. In addition, Yan X, et al. [41] also documented significant improvement in all domains of the NIH-CPSI, including the QoL domain, and in the QoL as per the AUA QoL_US.

A major strength of this study is that unlike most studies on the effect of ESWT on CP/CPPS, where the mean follow-up duration was 12 weeks (month 3) after ESWT, the present study followed patients up to 48 weeks (month 12) post-ESWT in order to rule out suggestions that the post-ESWT beneficial effects were transient or short-term. To the best of our knowledge, this is the longest documented follow-up duration for any study on the effect of ESWT in patients with CP/CPPS. Nevertheless, more studies exploring the long-term durability of ESWT efficacy and the safety profile across all standard clinical indices are warranted. Having said that, aside from one case of post-procedure dysesthesia, which was transient and mild, our results and observations indicate that ESWT is a safe treatment for CP/CPPS, as follow-up was uneventful, with no aggravated complications recorded through the entire 48 weeks of follow-up. None of the participants opted out of the study due to any reported treatment-related complication. Consistent with contemporary knowledge and documented reports, long-term complications of ESWT are unknown.

Like many studies of this nature, the present study has some limitations, including being a single-center study, thus prone to being critiqued for lack of external validation or the scientific rigor necessary for widespread generalization or consensus. Secondly, this was a prospective, single-arm cohort study, thus lacking a control or sham group for comparison and exclusion of placebo effect. Thirdly, the cohort size of 215 patients with CP/CPPS, though greater than the minimum necessary number (i.e., given an expected average improvement in CPSI total score of 5 points, the sample size required was 14 (α = 0.05, β = 0.8, σ = 6)) to meet the required statistical constraints, was relatively small and carried the risk of not representing CP/CPPS of all known pathoetiologies, thus necessitating the evaluation of the efficacy of ESWT in larger and multi-center cohort studies.

## 5. Conclusions

As summarized in our schematic abstract (Figure 3), the present study demonstrated that ESWT, an outpatient and easy-to-perform, minimally invasive procedure, effectively alleviates pain, improves erectile function, and ameliorates quality of life in patients with CP/CPPS. Our study highlighted the putative ability of ESWT to reverse the pathophysiology of CP/CPPS at the cellular level, elicit durable improvement in patients’ clinical status, and restore spontaneous erectile function, with minimal or null side effects.

## Figures and Tables

**Figure 1 jcm-10-03602-f001:**
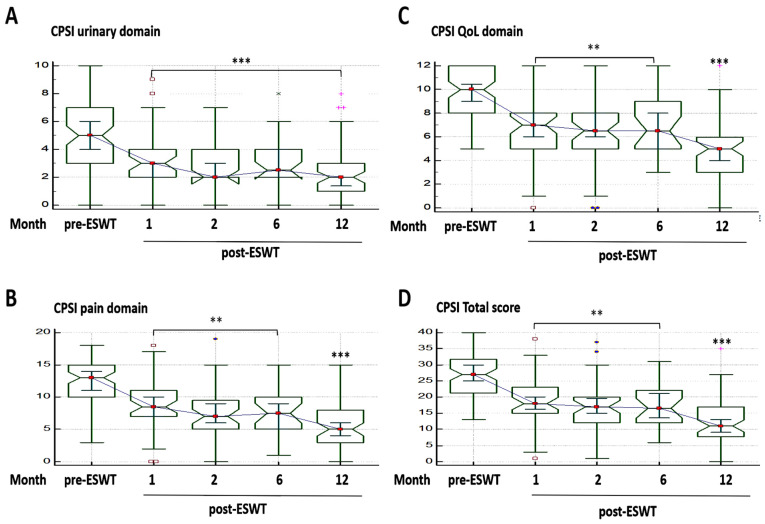
Extracorporeal Shockwave Therapy and Chronic Prostatitis Symptom Index (CPSI). Notched box-and-whiskers graphs showing the time-phased effect of extracorporeal shockwave therapy using the (**A**) urinary domain, (**B**) pain domain, (**C**) quality of life, and (**D**) total score over a period of 12 months. ** *p* < 0.01, *** *p* < 0.001.

**Figure 2 jcm-10-03602-f002:**
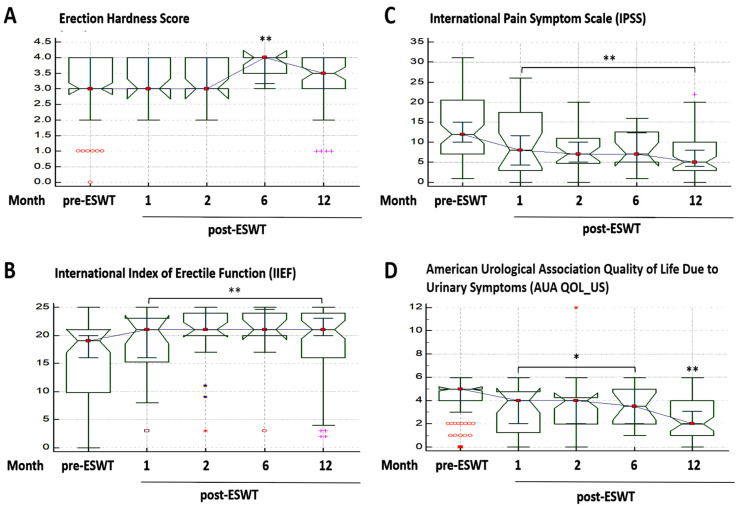
Effect of extracorporeal shockwave therapy in patients with chronic prostatitis/chronic pelvic pain syndrome (CP/CPSS). Notched box-and-whiskers graphs showing the time-phased effect of extracorporeal shockwave therapy on the (**A**) erection hardness score, (**B**) international index of erectile function, (**C**) international pain symptom scale, and (**D**) American Urological Association Quality of Life due to Urinary Symptoms over a period of 12 months. * *p* < 0.05, ** *p* < 0.01.

**Figure 3 jcm-10-03602-f003:**
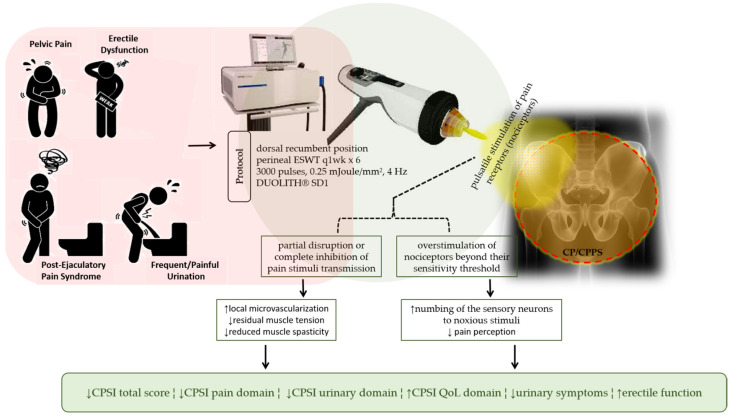
Schematic abstract: By disrupting pain stimuli transmission or overstimulation of nociceptors, ESWT effectively alleviates pain, improves erectile function, and ameliorates quality of life in patients with CP/CPPS through increased re-perfusion and numbing of sensory neurons to noxious stimuli, with associated reduction in residual muscle tension, spasticity, and pain perception.

**Table 1 jcm-10-03602-t001:** Comparison of paired samples parameters over time.

Pair		*n*	Variable 1 (Mean ± SD)	Variable 2 (Mean ± SD)	Paired Differences		
Variable 1	Variable 2				Mean ± SD	95% CI	*p*-Value ^a^
CPSI__urinary_domain_1	CPSI__urinary_domain_pESWT_1	41	4.32 ± 2.74	3.15 ± 2.17	−1.17 ± 1.87	−1.76 to −0.58	0.0003
CPSI__urinary_domain_1	CPSI__urinary_domain_pESWT_2	39	4.62 ± 2.75	2.82 ± 2.02	−1.79 ± 2.38	−2.56 to −1.03	<0.0001
CPSI__urinary_domain_1	CPSI__urinary_domain_pESWT_6	24	4.88 ± 2.63	2.88 ± 1.90	−2.00 ± 2.11	−2.90 to −1.11	0.0001
CPSI__urinary_domain_1	CPSI__urinary_domain_pESWT_12	59	4.85 ± 2.70	2.20 ± 1.92	−2.64 ± 2.66	−3.34 to −1.95	<0.0001
CPSI_pain__domain_1	CPSI_pain__domain_pESWT_1	41	11.85 ± 3.40	8.56 ± 4.30	−3.29 ± 3.72	−4.47 to −2.12	<0.0001
CPSI_pain__domain_1	CPSI_pain__domain_pESWT_2	39	12.36 ± 3.10	7.67 ± 4.23	−4.69 ± 4.40	−6.12 to −3.27	<0.0001
CPSI_pain__domain_1	CPSI_pain__domain_pESWT_6	24	12.29 ± 3.17	7.42 ± 3.86	−4.87 ± 4.01	−6.57 to −3.18	<0.0001
CPSI_pain__domain_1	CPSI_pain__domain_pESWT_12	59	12.29 ± 3.39	5.36 ± 3.62	−6.93 ± 4.45	−8.09 to −5.77	<0.0001
CPSI_QoL_domain_1	CPSI_QoL_domain_pESWT_1	41	9.27 ± 2.15	6.88 ± 2.87	−2.39 ± 2.96	−3.32 to −1.46	<0.0001
CPSI_QoL_domain_1	CPSI_QoL_domain_pESWT_2	39	9.51 ± 2.09	6.46 ± 2.96	−3.05 ± 3.15	−4.07 to −2.03	<0.0001
CPSI_QoL_domain_1	CPSI_QoL_domain_pESWT_6	24	9.42 ± 2.22	6.92 ± 2.80	−2.50 ± 2.47	−3.54 to −1.46	0.0001
CPSI_QoL_domain_1	CPSI_QoL_domain_pESWT_12	59	9.53 ± 2.05	4.85 ± 2.57	−4.68 ± 2.82	−5.41 to −3.94	<0.0001
CPSI_total_score_1	CPSI_total_score_pESWT_1	41	25.44 ± 6.96	18.59 ± 8.32	−6.85 ± 7.41	−9.19 to −4.52	<0.0001
CPSI_total_score_1	CPSI_total_score_pESWT_2	39	26.49 ± 6.77	16.95 ± 8.10	−9.54 ± 9.01	−12.46 to −6.62	<0.0001
CPSI_total_score_1	CPSI_total_score_pESWT_6	24	26.58 ± 7.03	17.21 ± 7.39	−9.38 ± 7.12	−12.38 to −6.37	<0.0001
CPSI_total_score_1	CPSI_total_score_pESWT_12	59	26.66 ± 6.95	12.37 ± 7.24	−14.29 ± 8.61	−16.53 to −12.04	<0.0001
EHS_1	EHS_pESWT_1	22	2.95 ± 1.17	3.36 ± 0.66	0.41 ± 0.67	0.11 to 0.70	0.009
EHS_1	EHS_pESWT_2	22	3.23 ± 0.97	3.45 ± 0.60	0.23 ± 0.61	−0.04 to 0.50	0.0961
EHS_1	EHS_pESWT_6	8	3.38 ± 0.74	3.63 ± 0.52	0.25 ± 0.71	−0.34 to 0.84	0.3506
EHS_1	EHS_pESWT_12	49	3.20 ± 0.82	3.31 ± 0.82	0.10 ± 0.68	−0.09 to 0.30	0.3019
IIEF_1	IIEF_pESWT_1	22	16.00 ± 7.89	18.36 ± 6.48	2.36 ± 3.13	0.98 to 3.75	0.0019
IIEF_1	IIEF_pESWT_2	22	18.14 ± 6.81	20.14 ± 5.77	2.00 ± 2.96	0.69 to 3.31	0.0046
IIEF_1	IIEF_pESWT_6	8	15.88 ± 7.86	19.50 ± 7.17	3.63 ± 3.93	0.34 to 6.91	0.0348
IIEF_1	IIEF_pESWT_12	50	17.06 ± 6.60	18.80 ± 6.53	1.74 ± 3.06	0.87 to 2.61	0.0002
IPSS_1	IPSS_pESWT_1	27	13.59 ± 8.31	10.37 ± 8.39	−3.22 ± 5.06	−5.23 to −1.22	0.0028
IPSS_1	IPSS_pESWT_2	27	13.81 ± 7.92	8.85 ± 5.95	−4.96 ± 5.99	−7.33 to −2.59	0.0002
IPSS_1	IPSS_pESWT_6	14	16.79 ± 9.21	7.93 ± 4.57	−8.86 ± 6.50	−12.61 to −5.10	0.0002
IPSS_1	IPSS_pESWT_12	56	13.71 ± 8.46	6.88 ± 5.14	−6.84 ± 6.29	−8.52 to −5.15	<0.0001
QoL_1	QoL_pESWT_1	26	4.04 ± 1.66	3.35 ± 1.92	−0.69 ± 1.69	−1.38 to −0.01	0.0473
QoL_1	QoL_pESWT_2	27	4.37 ± 1.42	3.48 ± 2.41	−0.89 ± 2.06	−1.71 to −0.07	0.0339
QoL_1	QoL_pESWT_6	14	5.14 ± 0.95	3.21 ± 1.72	−1.93 ± 1.33	−2.70 to −1.16	0.0001
QoL_1	QoL_pESWT_12	56	4.25 ± 1.59	2.61 ± 1.57	−1.64 ± 1.59	−2.07 to −1.22	<0.0001

^a^: Paired samples t-test; CPSI/NIH-CPSI = National Institute of Health Chronic Prostatitis Symptom Index; 95% CI = 955 confidence interval; SD = standard deviation; VAS = visual analog scale; IPSS = International Prostate Symptom Score; QoL/AUA QoL_US = American Urological Association Quality of Life Due to Urinary Symptoms; IIEF = International Index of Erectile Function; EHS = erectile hardness score; ESWT = extracorporeal shockwave therapy; pESWT = post-extracorporeal shockwave therapy.

**Table 2 jcm-10-03602-t002:** Review of previous studies on ESWT in patients with CP/CPPS.

Study	Study Design	No. of Patients	Baseline NIH-CPSI Score	Intervention: ESWT	Treatment Duration	Follow-Up (Weeks)	Outcome (at the End of Follow-Up)
Rayegani 2020	RCT	31	27.87 ± 7.2	4 sessions of focused ESWT (a protocol of 3000 impulses, 0.25 mJ/mm^2^ and 3 Hz of frequency)	Once a week for 4 weeks	1, 4, 12	NIH-CPSI (↓), VAS (↓), Qmax (↑), PVR (↓), IPSS (↓), IIEF (↓), NIH QOL (↑)
Zhang 2019	Non-RCT	45	28.52 ± 4.07	rESWT (3000 pulses each; pressure: 1.8–2.0 bar; frequency: 10 Hz)	Once a week for 8 weeks	1, 4, 8, 12	NIH-CPSI (↓), VAS (↓), IPSS (↓), IIEF (↑), NIH QOL (↓)
Guu 2018	Cohort	33	28.03 ± 6.18	3000 impulses at a frequency of 4 Hz, with a energy density of 0.25 mJ/mm^2^	Once a week for 4 weeks	1, 4, 12	NIH-CPSI (↓), VAS (↓), IPSS (↓), IIEF-5 (↑), EHS(−), IELT(−)
Salecha 2017	Cohort	50	NA	2500 impulses	Once a week for 4 weeks	1, 4, 12	NIH-CPSI, VAS (↓), ultrasound, PSA level
Letizia 2017	Cohort	39	NA	NA	Once a week for 6 weeks	1, 6, 12	pain score, urinary score, quality-of-life (NIH-CPSI?)
Al Edwan 2017 (1 year follow up of Mohammad 2016?)	Cohort	41	27.7 ± 7.6	2500 impulses at a frequency of 3 Hz, with a energy density of 0.25 mJ/mm^2^	Once a week for 4 weeks	2, 6 months, 12 months	NIH-CPSI (↓), IPSS (↓), AUA QOL_US (↓), IIEF (↑)
Turcan 2016	Cohort	20	NA	Frequency of 8 Hz	4 times weekly for ?	4, 26	NIH-CPSI
Pajovic 2016	RCT	30	31.06 ± 7.75	3000 impulses at a frequency of 3 Hz, with a energy density of 0.25 mJ/mm^2^	Once a week for 4 weeks	12, 24	NIH-CPSI (↓), ultrasound
Mohammad 2016	Cohort	25	NA	2500 impulses over 13 min	Once a week for 4 weeks	2	NIH-CPSI (↓), IPSS (↓), AUA QOL_US (↓), IIEF (↑)
Kulchavenya 2016	Cohort	27	NA	2000–3000 impulses with a energy density of 0.056-0.085 mJ/mm^2^	Twice weekly for 3 weeks	1, 4	NIH-CPSI (↓), LDF
Moayednia 2014	RCT	19	26.03 ± 3.72	3000 impulses at a frequency of 3 Hz, with a energy density of 0.25 mJ/mm^2^	Once a week for 4 weeks	16, 20, 24	NIH-CPSI(−), VAS(−)
Vahdatpour 2013	RCT	40	26.5 ± 3.4	3000 impulses at a frequency of 3 Hz, with a energy density of 0.25–0.4 mJ/mm^2^	Once a week for 4 weeks	1, 2, 3, 12	NIH-CPSI(−), VAS(?)
Kernesiuk 2013	Cohort	15	NA	NA	Once a week for 4 weeks	1, 2, 4, 12	NIH-CPSI(↓in QOL and pain domain)
Zeng 2012	RCT	40	30.5 ± 4.7	2000 impulses at a frequency of 2 Hz, with a energy density of 0.06 mJ/mm^2^- max tolerated dose	5 times weekly for 2 weeks	4, 12	NIH-CPSI (↓)
Mathers 2011	Cohort	14	26.1 ± 1.8	NA	Once a week for at least 3 weeks	4, 12	NIH-CPSI (↓)
Zimmermann 2010 (1 year follow up Zimmermann 2009)	RCT	44	NA	3000 impulses at a frequency of 3 Hz, with a energy density of 0.25 mJ/mm^2^		1, 3, 6, 12 months	NIH-CPSI, VAS, IPSS, IIEF
Zimmermann 2009	RCT	30	23.20 ± 0.66	3000 impulses at a frequency of 3 Hz, with a energy density of 0.25 mJ/mm^2^	Once a week for 4 weeks	1, 4, 12	NIH-CPSI (↓), VAS (↓), IPSS (↓), IIEF (↑)
Zimmermann 2008	Cohort Study	14 20	10.0 19.9	2000 impulses at a frequency of 3 Hz, with a energy density of 0.11 mJ/mm^2^ 3000 impulses at a frequency of 3 Hz, with a energy density of 0.25 mJ/mm^2^	3 times weekly for 2 weekOnce a week for 4 weeks	1, 4, 12 1, 4, 12	NIH-CPSI, VAS, IPS SNIH-CPSI (↓), VAS (↓), IPSS(−)

ESWT: extracorporeal shock wave therapy, rESWT: radial extracorporeal shock wave therapy; CP/CPPS: chronic pain/chronic pelvic pain syndrome, NIH-CPSI: national institute of health-chronic prostatitis symptom index, VAS: visual analogue scale, IIEF-5: 5-item version of the international index of erectile function, EHS: erection hardness score, IELT: intravaginal ejaculation latency time, AUA QOL_US: American urological association quality of life due to urinary symptoms, Qmax: maxium flow rate; PVR: post-void residual urine; LDF: laser Doppler flowmetry, (↓): statistical significance decrease (*p* < 0.05), (↑): statistical significance increase (*p* < 0.05), (−): no statistical difference (*p* > 0.05), NA: not available. Question mark (?) implies lack of certainty, as the cited study itself lacked clarity on the association.

## Data Availability

The data used and analyzed in the current study are available on request from the corresponding author.

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
