# Peer review of "Extracorporeal Shockwave Therapy (ESWT) Alleviates Pain, Enhances Erectile Function and Improves Quality of Life in Patients with Chronic Prostatitis/Chronic Pelvic Pain Syndrome"

_jcm, 2021, doi:10.3390/jcm10163602_

Round 1

Reviewer 1 Report

In this study, the authors showed the outcomes of extracorporeal shockwave therapy (ESWT) for chronic prostatitis/chronic pelvic pain syndrome (CP/CPPS). Although many patients suffer from CP/CPPS, no effective treatment has been established to date. This is the single-institutional, largest-ever study of ESWT for CP/CPPS and may help in future CP/CPPS treatment.  I feel the data is interesting, but several revisions is needed .

1. The introduction is too long. General information about CP/CPPS should be shortened.

2. In the result section, how many patients completed the 1-year evaluations? It is particularly important when unsatisfactory patients were more likely to drop out of the study. 

3. If some information are lacking, how to treat or impute the missing data? it should be described in the Method.

4. Also, how many patients received other therapies, such as medical therapy/surgery? Did the results include the patients who underwent TURP during follow-up? The follow-up data should be described.

5. how many patients received antibiotics for C. trichomatis or T. Vaginalis?

6. Table 1 and  Figure1/2 showed the same data. Either one is fine.

7. The sample size calculation did not make sense. Why prostate cancer mortality was estimated?

Reviewer 2 Report

This study shows that CP/CPPS patients were treated with extracorporeal shockwave therapy and maintained their effectiveness for a relatively long period of time. Although this study is valuable, please consider corrections to the comments below.

1. Please describe the method of shock wave to the prostate in a little more detail. Are you taking suppositories before treatment? Please explain in a way that is easy for the reader to understand, such as how to set the machine (using a schema, etc.).

2. Please comment on the mechanism by which shock wave treatment reduces prostate pain.

3. Please describe the presence or absence of complications in the patient who received the shock wave. Please also describe other disadvantages of shock waves.

4. Is shock wave treatment provided by insurance in your country?

5. How different do you think the effectiveness of shock wave intensity (low or high) ? Also, please mention how the method the authors are doing differs from other treatises.

Round 2

Reviewer 1 Report

The authors appropriately answered the question I raised except for  Q7.

What they calculated was the sample size needed to estimate the percentage of people who have CPPS.

Since this is an observational study, I recommend the authors delete the sample size estimation and Supple Fig 1.

If not, I recommend the authors clarify what they want to estimate. ( i.e. Given an expected average improvement in CPSI total score of 5 points, the sample size required is 14 [α = 0.05, β = 0.8, σ = 6].) Please contact your statistician.

Reviewer 2 Report

The authors answered all the questions in good faith. I have another question about the extracorporeal shock wave therapy device. Please add the description that this machine is fixed to something or is carried by hand during treatment.
